# School closures significantly reduced arrests of black and latinx urban youth

**Jessica T. Simes**[1], **Tori L. Cowger**[2], **Jaquelyn L. Jahn**[3]*

1 Department of Sociology, Boston University, Boston, MA, United States of America, 2 FXB Center for Health and Human Rights, Harvard University T.H. Chan School of Public Health, Boston, MA, United States of America, 3 The Ubuntu Center on Racism, Global Movements, and Population Health Equity, Drexel University Dornsife School of Public Health, Philadelphia, PA, United States of America

* jj3222@drexel.edu

## Abstract

### Background & methods

Youth of color are surveilled and arrested by police at higher rates than their White peers, contributing to racial inequities across the life course and in population health. Previous research points to schools as an increasingly relevant site for youth criminalization, but existing studies emphasize within-school mechanisms, with limited analysis of policing in surrounding school areas. To fill this gap, we study changes in police arrests of youth after initial COVID-19 school closures in 2020 across four US cities overall and in relation to public school locations. We analyze geocoded arrest records and use interrupted time series negative binomial regression models with city and month fixed effects to estimate change in weekly arrest rates among White, Black, and Latinx youth. We estimate arrest densities within school areas before and after school closures using spatial buffers of 300 feet.

### Results

In the immediate weeks and months following COVID-19 pandemic school closures, youth arrest rates fell dramatically and with lasting impacts compared to other age groups. During the period of remote learning, weekly youth arrest rates declined by 54.0% compared to youth arrests rates in 2019, adjusting for city and seasonality (Incident Rate Ratio 0.46, 95% CI: 0.41, 0.52). We estimate Black youth weekly arrests fell from 43.6 to 16.8 per 100,000, vs. 4.6 to 2.2 per 100,000 among White youth. However, Black youth arrest rates during the remote learning period were still nearly 5 times that of White youth pre-pandemic. We also find that youth arrest rates declined during two school closure periods: at the start of the pandemic in 2020 and during Summer 2019. A spatial analysis shows Black and Latinx youth arrest densities in the surrounding 300 feet of K-12 schools were at least 15 and 8.5 times that of White youth, respectively, in both pre- and remote-learning periods.

### Conclusions

Black and Latinx youth face a higher likelihood of being arrested near a school than do White youth and older age groups, and racial inequities in arrests remains after school

**Data Availability Statement:** Data are available on Dataverse https://doi.org/10.7910/DVN/46Q4HF.

**Funding:** This study was funded by Social Science Research Council Rapid-Response Grants on COVID (JTS, JLJ), Scholars Strategy Network

(JTS, JLJ), and Boston University Initiative on Cities (JTS, JLJ). The funders had no role in study design, data collection and analysis, decision to publish, or preparation of the manuscript.

**Competing interests:** The authors have declared that no competing interests exist.

closures. Our findings show school closures significantly reduced arrests of urban youth of color, and policies addressing youth criminalization and structural racism should consider the joint spatial context of schools and policing. Although school closures may have resulted in learning loss and harms to youth wellbeing, closures interrupted comparatively high levels of arrest for Black and Latinx youth.

## Introduction

Police encounters and arrests substantially impact health and life chances [1–4]. Coupled with the comparatively high prevalence of policing in communities of color, there is broad national interest in policing reform as one way to address structural racism in the United States. While research and policy related to policing have largely focused on US adult populations, most first encounters with police happen during childhood [5]. By the time youth reach young adulthood, approximately 25–41% will have been arrested at least once, and nearly half of all Black boys will have been arrested by age 23 [6, 7].

A main way that youth (defined here as children and adolescents aged <18 years) encounter police is at school. In the most recent available data, schools in the US reported over 290,600 students to law enforcement during the 2015–2016 school year, disproportionately reporting Black students [8]. The extent to which schools are a site of criminalization for youth of color has generated a large body of research. This research mainly focuses on within-school dynamics that constitute what scholars have termed the "school-to-prison pipeline," including the rise of school-based police officers, criminalizing labels and punishments (e.g., "at-risk", suspension/expulsion), and criminal charges (e.g., truancy) [9–14]. From metal detectors to locker searches to harsh discipline, scholars stress that the "long reach of the carceral state" has suffused classrooms and the broader educational system with a culture of control [15–18].

Beyond within-school policies and practices that can lead to youth arrests, the areas surrounding schools are potentially important sites for surveillance and criminalization. While traditional policing strategies focus on large geographic areas, more recent policing strategies focus on much smaller spatial areas, targeting specific addresses, street corners, or blocks [19, 20]. Schools can be targeted police surveillance, stops, and arrests in the hours surrounding the school day [21–23]. For example, the "Hot Spots Policing Made Easy" chapter of a 2019 textbook for police practitioners and researchers suggests that police should take a targeted approach by patrolling the area around a given local high school "before or after school, when kids are either walking to school or walking home" [24]. Indeed, attending school fundamentally structures the routine activities of youth, exposing them to neighborhood contexts and transportation methods where they may encounter police as they travel to and from school [10]. In addition to police practices, state and local lawmakers established drug, gun, and gang free school zones and zero-tolerance policies, deepening the spatial link between schools and criminal law enforcement [25–27].

Ethnographic research points to how schools and other non-carceral institutions are important sites for youth-police contact and arrests. Schools have been analyzed as part of a "youth control complex," a system of "wraparound incarceration," or a "carceral continuum," where interactions with police across home and school contexts result in the criminalization of youth of color and affect their orientation towards both schooling and the criminal justice system [15, 28, 29]. For example, high school students interviewed by Carla Shedd described police treating Black youth differently from White youth after school, where Black students were told

to "clear the corner" while standing at the bus stop across the street from their school, ignoring groups of White students nearby [15]. More generally, urban researchers emphasize the importance of studying school-based inequalities as embedded in a larger system of residential segregation [30], which may be reinforced by police practices [31, 32].

The broader spatial context of schools is thus an underexamined mechanism explaining rates and inequities in arrests of youth. Moreover, we lack estimates of the prevalence of police arrests by age and race/ethnicity in the areas surrounding schools. In this analysis, we examine how arrests of urban youth changed after school closures, both in terms of overall rates and inequities, as well as the geographic proximity of arrests to schools. Like the school day, school closures may temporarily deemphasize schools for police surveillance and activity. As schools closed for several months during the early months of the COVID-19 pandemic, changes to the rates and location of youth arrests during this period are informative of the role that schools play in the arrests of youth as compared to other urban spaces, other age groups, and racial/ethnic inequities. Using the COVID-19 pandemic as a period of dramatic change to both in-person school attendance and policing practices, our overall aim is to estimate the spatial and demographic patterning of youth arrests in relation to school location. Thus, our paper asks: Given evidence of harmful effects of youth police contact and racial disparity, how does proximity to a school influence youth exposure to police arrest, particularly for Black and Latinx youth?

This paper provides estimates of arrest rates for youth before and after school closures during the first 9 months of the COVID-19 pandemic across Boston, Charleston, New York City, and Pittsburgh. We use 2020 COVID-19 pandemic school closures as a novel exogenous interruption to in-school attendance and thus potentially youth-police contact in school areas. Using interrupted time series negative binomial regression models with city and month fixed effects, we estimate change in weekly arrest rates among White, Black, and Latinx youth. To examine changes in arrests of youth in the context of COVID-19 school closures, we contrasted youth arrest rates from the pre-closure period (January 2019 through mid-March 2020) with two post-closure periods: 1) remote learning and 2) partial return (either hybrid or phased in in-person learning) to the classroom in two cities. Next, we contrast arrest rates in post-closure periods to another school closure period: Summer 2019. While we are interested in the impact of school closures on youth criminalization in a general sense, patterns of youth arrests during the early months of the COVID-19 pandemic in the US are substantively important for understanding potential drivers of health disparities, as arrests and incarceration are associated with greater risk of SARS-CoV-2 transmission [33]. Declines in arrests may also have implications for youth social and economic outcomes, including education, employment, and future criminal justice contact [34]. Finally, we contrast overall youth arrest rates and the spatial density of arrests surrounding K-12 public school locations to other age groups for whom we would not expect arrests to cluster around schools but would be similarly affected by pandemic stay-at-home orders. We estimate school areas using spatial buffers of 300 feet and use bootstrap resampling methods to provide uncertainty intervals for our spatial estimates. We analyze changes to arrests for all other age groups, within cities, and using multiple buffer densities in the Supporting Information (SI). We describe the data in Materials and Methods.

We estimate that after remote learning was implemented, youth arrest rates declined by 54% on average across the four cities. Rather than this being limited to pandemic-related school closures, we find comparable declines in youth arrest rates during Summer 2019 particularly within two cities and among youth of color. While all age and race/ethnicity groups saw significant declines in arrests following urban stay-at-home orders and school closures, Black and Latinx youth were significantly more likely than any other race/ethnicity and age group to experience a decline in arrests within surrounding school areas. The magnitude of the absolute

change in arrests for Black and Latinx youth suggests that school closures had population-level effects for reducing criminalization of urban youth of color by deemphasizing school areas for police activity. However, despite these steep and sustained declines among Black and Latinx youth, arrest rates never approach those observed among White youth, even in comparison to pre-school closure rates. These findings contribute to a body of research that explores both the level and inequity in youth criminalization in the United States by studying how both time and space in relation to schooling and police practice may increase risk for youth criminalization.

## Materials and methods

### Data

We study anonymized arrest records provided by police departments in four US cities: Boston, Charleston, New York City, and Pittsburgh. The Boston Police Department provided researchers with arrest data upon request, but data from other cities is publicly available from city police department websites. We selected these cities based on a search of publicly available data on geocoded arrests that covered all months from 2019 through 2020. All four cities enacted primary and secondary school closure orders during March 2020 and varied in terms of reopening schools during the fall of 2020.

All arrest data contain geographic (XY) data on arrest location from January 1, 2019 to December 31, 2020. Arrest records also include the age, gender, and race/ethnicity of the individual arrested. We geocoded the XY arrest data to each of the four cities and linked these data to the American Community Survey Five-Year Estimates (2015–2019), which provided data on city-level populations to construct arrest rates by race/ethnicity and age. We thus analyze arrest data for four cities observed across 104 weeks (2019–2020). We observe 390,526 arrests in the two years of data; only 434 arrests (0.1% of data) are missing race/ethnicity and/or age information and are removed from the analysis. Note that New York City provided age categories rather than exact age, so we can only identify youth as those categorized by the cities as being under 18. For all analyses we use the following age categories: under 18, 18–24, 25–64, and 65 and over; we use the following race/ethnicity categories provided by police departments: Black, White, and Latinx. Charleston does not identify ethnicity in their data, so results for Latinx people are limited to Boston, New York City, and Pittsburgh.

Administrative data are well-suited for studying the youth arrest rates and racial disparities therein. Surveys rely on participants' responses to questions asking if they have been arrested since the last interview [35]. When the criterion validity of survey responses can be assessed against an official record, researchers find that arrests are underreported in surveys generally, but that Black survey participants are more likely to under-report police contact than White participants, attenuating estimated racial disparities [36–38]. While administrative records from police departments lack data on other individual characteristics beyond a limited conceptualization of race, ethnicity, age, and gender categories, we believe these data are useful for a study of arrest prevalence, time trends, and racial disparities.

To identify school closure and reopening policies, we used data from the Center on Reinventing Public Education's School District Responses to COVID-19 Closures Database [39]. To assess the impact of spatial proximity to schools on arrests of youth, we also collected data on the location of K-12 public schools in each of the four cities. This publicly available geocoded public-school dataset contains all public elementary and secondary education facilities as defined by the Common Core of Data, National Center for Education Statistics, and US Department of Education for the 2017–2018 school year.

## Statistical analysis

To examine changes in arrests of youth in the context of COVID-19 school closures, we contrasted youth arrest from the pre-closure period (January 2019 through mid-March 2020) with two post-closure periods: 1) the remote learning period and 2) the period in which some students returned to the classroom in New York City and Charleston in September-December 2020 (students in Boston and Pittsburgh remained remote during this time and thus do not contribute data to the partial return period, only to the remote learning period). As a count of arrests within cities, we estimated interrupted time series negative binomial regression models with a log population offset to generate average weekly log rates of age-specific arrests in the remote and return periods as compared with the pre-period. We chose negative binomial regression due to significant overdispersion in the dependent variable, and Akaike Information Criteria indicated a negative binomial regression approach had a better model fit compared to Poisson models. The expected arrest rate $Y_{it}$, can be written in a negative binomial regression:

$$\log(Y_{it}) = \beta_0 + \beta_{1m} + \beta_{2i} + \delta_p,$$

where $i$ is for city, $t$ is for week, $m$ is for month, and $p$ is for pandemic period. $\beta_{2i}$ are city fixed effects to account for all time-invariant city-specific confounders (including overall population size and density), and $\beta_{1m}$ are month fixed effects to address seasonality. $\delta_p$ estimates the average change in arrest rate during remote learning and, when applicable, partial return to in-person teaching, compared to the pre-closure period. We report incident rate ratios (IRR), which is the ratio of arrest rates in each post period to the number of arrests in the pre-period, with 95% confidence intervals. Confidence intervals in all models were calculated using robust standard errors to account for clustering within cities. We repeated this model for overall youth arrest rates within cities (S3 Fig) and by race/ethnicity (S4 Fig).

To assess the impact of spatial proximity to schools on youth criminalization, we compared the density of arrests by age group and race/ethnicity in areas immediately surrounding schools. We calculated the percentage of total arrests and density of arrests per km$^2$ occurring within a buffer with a radius of 300 feet (0.09 km) around each school. Thus, arrests within buffer zones would include any arrests made by city police officers on a school site and those beyond it (but within the buffer). This distance was adapted from Massachusetts' statute defining drug free school zones, and the shortest zone distance of all cities included in this study [25]. While these school areas represent approximately 3% of land area across included cities (ranging from 0.3% in Charleston to 4.2% in New York City), 11.3% of all youth arrests during pre- and remote-learning periods occurred in these areas (ranging from 6.6% in Boston to 19.5% in Charleston) (S3 Table, S5 Fig). We report point estimates and 95% confidence intervals from 500 iterations of bootstrap resampling stratified by city and age group. We contrast arrests of youth (under age 18) to young adults (age 18–24) to compare youth arrests to a proximate age group that would be less likely to be affected by school closures.

## Sensitivity analyses

We conducted three sensitivity analyses. First, we estimated a more conservative temporal comparison that set the pre-period to Summer 2019, when schools were closed for summer recess and arrest rates of youth were lowest. By contrast, our main models compare arrest rates after school closures to average rates in 2019 through early March 2020. Models for our more conservative temporal comparison included the same covariates and contrasted rates of youth arrests in Summer 2019 to rates in the remote and partial return periods. In separate sensitivity analyses not reported in this paper, we found arrest declines in Summer 2019 were strongest

among Black and Latinx youth. Second, in spatial analyses, we also considered school zone buffer distances of 1,000 feet (New York City and Pittsburgh) and 2,640 feet (Charleston) (S3 Table). We prioritize school zone buffer distances of 300 feet in our main analysis, as larger buffer distances cover a large proportion of city land area and youth arrests (S5 Fig, S3 Table). For example, in New York City, buffer distances of 1,000 feet cover 32.6% of the city's land area and 69.2% of youth arrests. Third, we compare changes in density of arrests per km$^2$ within school buffer zones to changes in arrest density in areas outside of school buffer areas (S9 Fig).

## Results

Table 1 describes demographic characteristics of the population of arrested youth (under age 18) and young adults (age 18–24)—a key age comparison group in our study—across the four cities. We observe 18,023 arrests of youth during the study period (2019–2020), where New York City alone accounts for 16,123 arrests. Across all cities, Black youth represent 26.4% of the youth population, but 63.5% of all youth arrests. Conversely, White youth represent 26.9% percent of all youth in these cities but account for 4.9% of youth arrests. Thirty four percent of the youth population were Latinx, and they represent 27.9% of youth arrests. The youngest people we observe in our sample are two 8-year-old children; a Black 8-year-old girl was arrested for shoplifting, the other, a Black 8-year-old boy, was arrested for unlawful carry of a weapon on school property. Both children were arrested in 2019 before the COVID-19 pandemic affected US cities. We observe similar patterns of racial and ethnic inequities in arrests among young adults in our sample (Table 1).

### Youth arrest rates and disparities before and after school closures

After COVID-19 school closures, youth arrests markedly declined in the four cities (Fig 1). In the weeks prior to each state's school closures (January 2019 –March 2020), average weekly youth arrest rates were 17.4 per 100,000, dropping to a weekly average of 7.12 per 100,000 after school closures. We also observed declines in arrests for other age groups in the period after school closures, which closely aligned to when stay-at-home orders were implemented in these cities (S1 Fig).

To compare pandemic school closures to another, more commonplace, school closure we analyzed youth arrest rates during school closures to the period in 2019 when rates were lowest. Youth arrests rates significantly decreased during Summer 2019, when schools were closed for summer recess, nearly approaching levels of arrests during the 2020 COVID-19 pandemic

**Table 1. Percentage distribution of youth and young adult arrested and total population in Boston, Charleston, New York City, and Pittsburgh, 2019–2020.**

|  | Black Arrests (%) | Black Pop (%) | Latinx Arrests (%) | Latinx Pop (%) | White Arrests (%) | White Pop (%) | Total arrests (N) | Total Pop (N) |
|---|---|---|---|---|---|---|---|---|
| *Age <18 years* | 63.54 | 26.41 | 27.87 | 33.96 | 4.87 | 26.98 | 18,023 | 1,931,699 |
| Boston | 65.56 | 36.40 | 28.01 | 31.87 | 5.60 | 24.29 | 482 | 109,003 |
| Charleston | 75.98 | 28.52 | – | 3.98 | 23.83 | 62.48 | 512 | 22,648 |
| New York City | 61.64 | 25.52 | 30.27 | 35.23 | 3.98 | 26.19 | 16,123 | 1,754,547 |
| Pittsburgh | 89.29 | 35.93 | 0.88 | 4.70 | 9.82 | 45.88 | 906 | 45,501 |
| *Age 18–24 years* | 52.52 | 24.90 | 32.78 | 30.23 | 8.85 | 31.47 | 78,935 | 898,451 |
| Boston | 55.77 | 19.58 | 28.42 | 19.37 | 14.37 | 46.51 | 1,879 | 101,601 |
| Charleston | 50.31 | 18.74 | – | 4.74 | 49.06 | 73.07 | 1,604 | 16,655 |
| New York City | 51.38 | 26.32 | 35.16 | 34.17 | 7.12 | 26.03 | 71,953 | 729,554 |
| Pittsburgh | 75.22 | 17.09 | 1.09 | 3.55 | 23.09 | 66.00 | 3,499 | 50,641 |

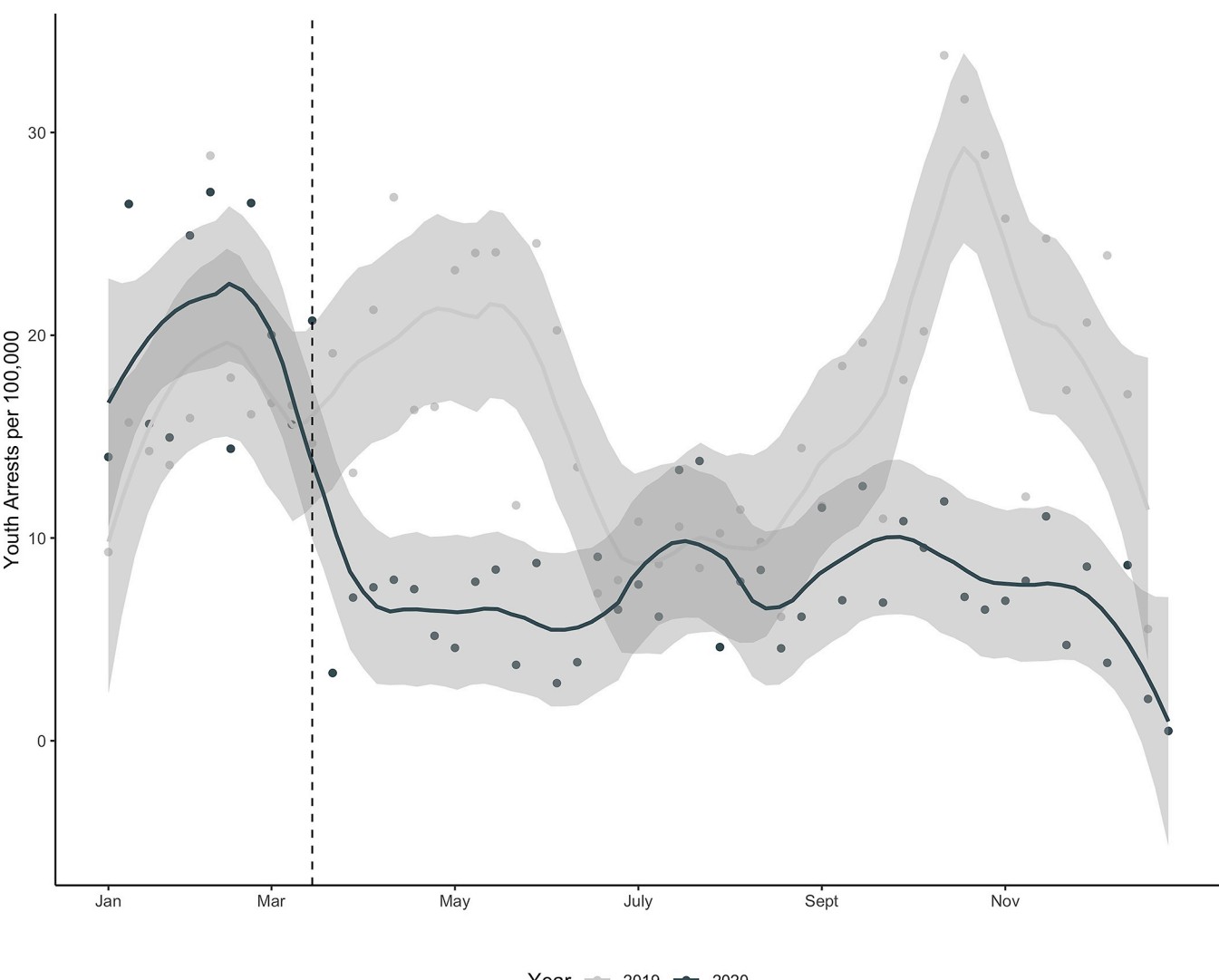

**Fig 1. Temporal trends in youth arrest rates in Boston, Charleston, New York City and Pittsburgh, 2019–2020.** Weekly averages smoothed using a loess function (span = 0.25).

summer months (see Fig 1). These results contrast with prior research suggesting that summer months have a higher probability of criminal activity for youth who are released from the routines and structure of school, which would partially be reflected in arrest levels. These results support the hypothesis that school closures (i.e., summer recess) reduce risk of arrest for youth as police temporarily de-emphasize schools for police activity. Stratifying by city, we found that in comparison to Summer 2019, there were lower rates of arrest during 2020 pandemic-related school closures in Boston and New York City but decreases in youth arrests in Charleston and Pittsburgh were not statistically significantly different from 2019 summer rates (S2 Fig).

Fig 2 displays results from interrupted time series negative binomial regression analyses predicting average weekly arrest rates after school closures across age groups, compared to rates in the year prior. These models test whether school closure periods were uniquely impactful for youth under 18, or if other age groups followed similar trends. After the start of

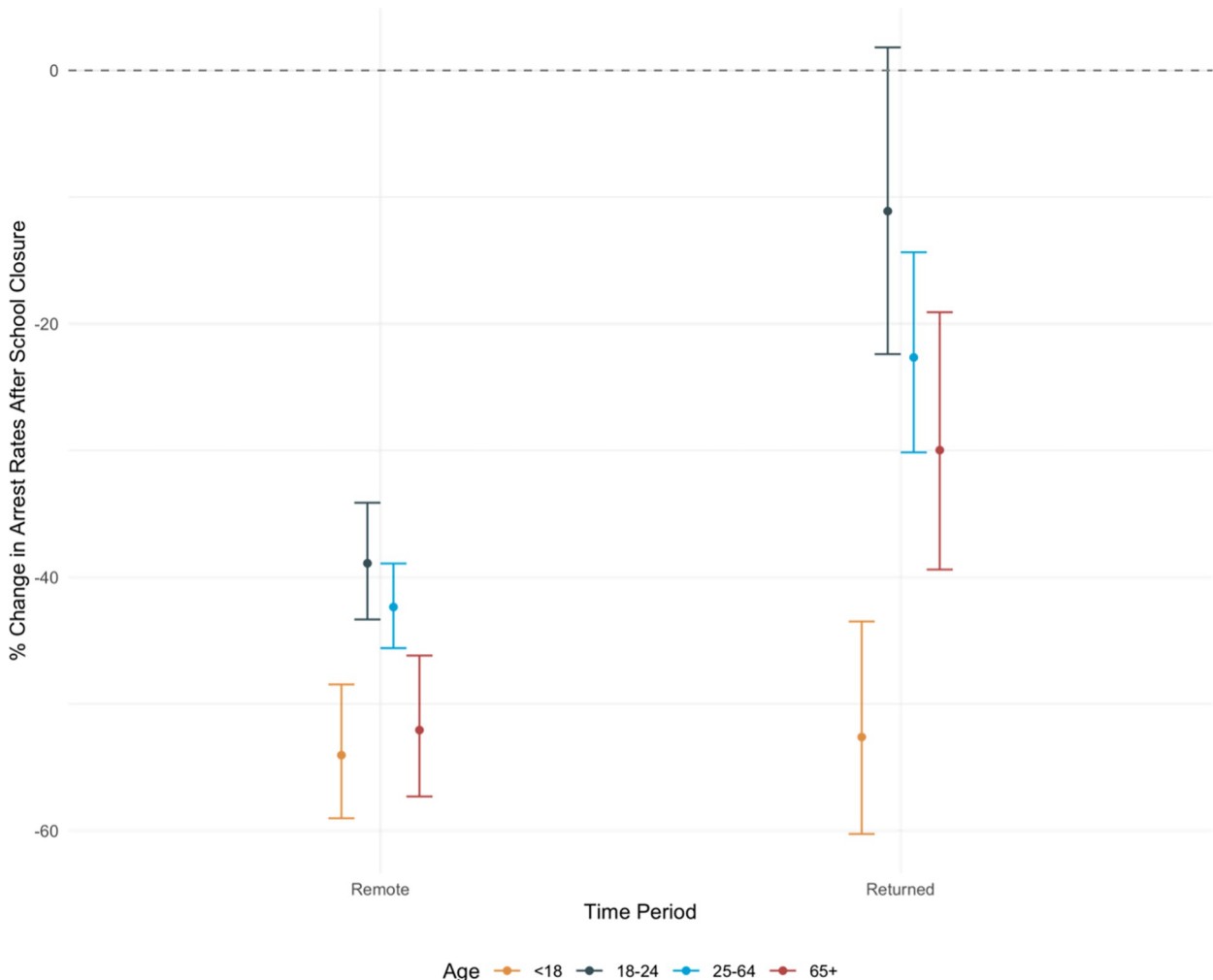

**Fig 2. Percentage change in rates of arrest by age group before vs. after school closures in Boston, Charleston, New York City, and Pittsburgh, 2019–2020.** Estimates from four age-specific negative binomial models predicting rates of arrests with a population offset and fixed effects for city and month. The referent time period for these models is 2019-March 2020. Only New York City and Charleston contribute data to the "Partially Returned" period because Boston and Pittsburgh remained remote after September 2020. Complete regression results reported in S1 Table.

the pandemic when remote learning was implemented, weekly arrests declined for all age groups (Fig 2). During this time, weekly youth arrest rates declined by 54.04% compared to youth arrests rates in 2019, adjusting for city and seasonality (IRR 0.46, 95% CI: 0.41, 0.52). For Charleston and New York City, which partially returned to in-person instruction in September 2020, weekly youth arrest rates during this return period declined by 52.6% relative to the year prior (IRR: 0.47, 95% CI: 0.40, 0.57). However, for other age categories, during the period between September and December 2020, declines in arrests relative to 2019 become less strong (Fig 2) and arrests of young adults are not significantly different than 2019 (IRR: 0.89, 95% CI: 0.78, 1.02). Relative rates of decline in weekly youth arrests were consistent across cities (S3 Fig). S1 Table reports the complete regression results displayed in Fig 2.

Arrest declines among youth are most starkly observed among Black youth for whom average weekly arrest rates dropped from 43.6 to 16.8 per 100,000 (Fig 3). By comparison, average weekly arrests of White youth declined from 4.6 to 2.2 per 100,000 (Fig 3). We report

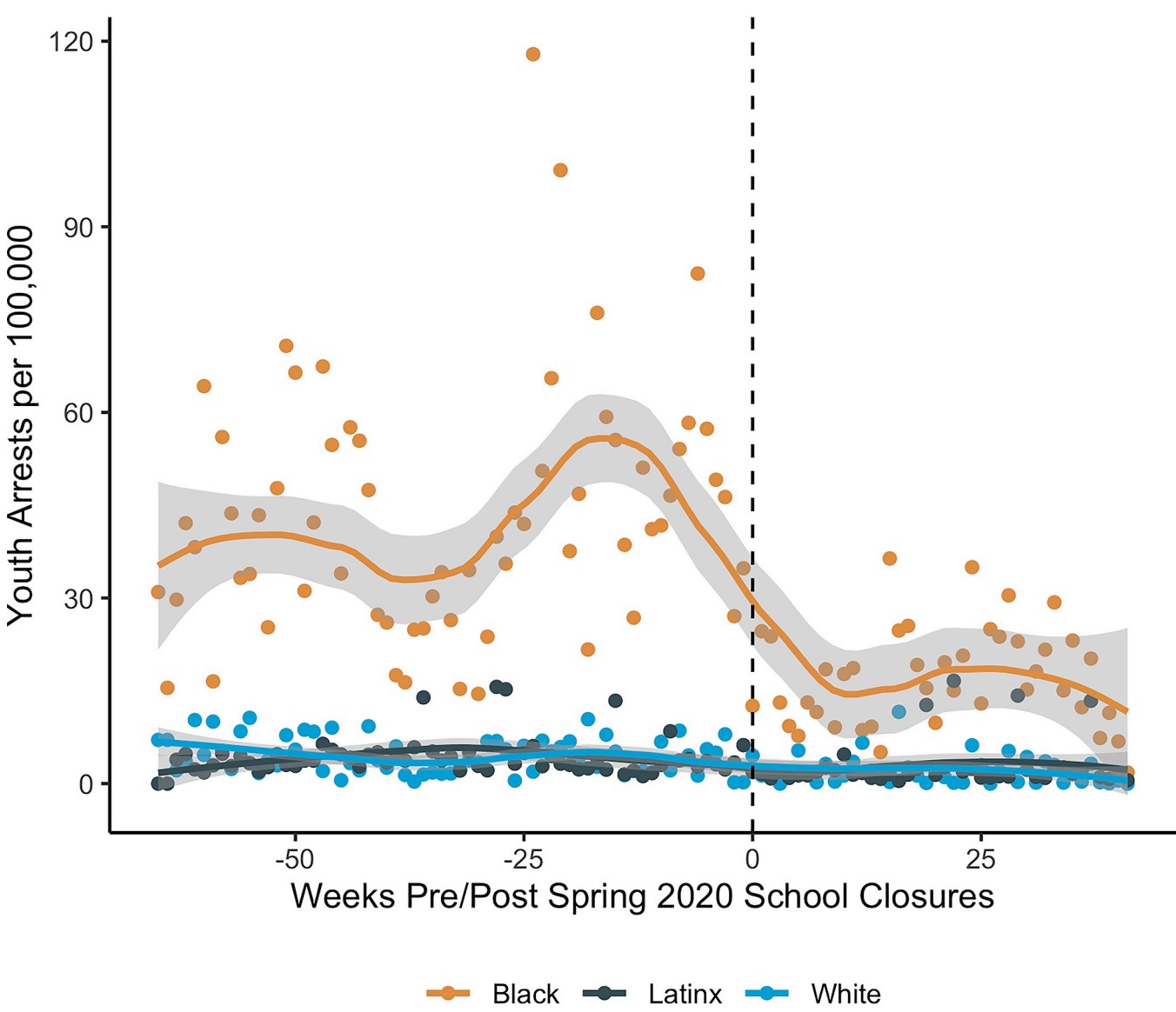

**Fig 3. Temporal trends in youth arrest rates by race in Boston, Charleston, New York City, and Pittsburgh, 2019–2020.** Weekly averages smoothed using a loess function (span = 0.5).

interrupted time series negative binomial regression results for youth arrests stratified by race/ethnicity, and find Black, White, and Latinx youth experienced significant percentage declines overall, and these declines were sustained throughout the remote-learning period and partial returns to in-person learning (S4 Fig). However, despite these steep percentage declines for all groups, weekly arrest rates for Black youth never approach rates observed among other racial/ethnic groups, even in comparison to rates observed before closures (Fig 3).

## The spatial pattern of school locations and youth arrest before and after COVID-19 school closures

Fig 4 shows average weekly arrest densities among youth in New York City school buffer areas and surrounding city areas during pre- and remote-learning periods (other cities shown in

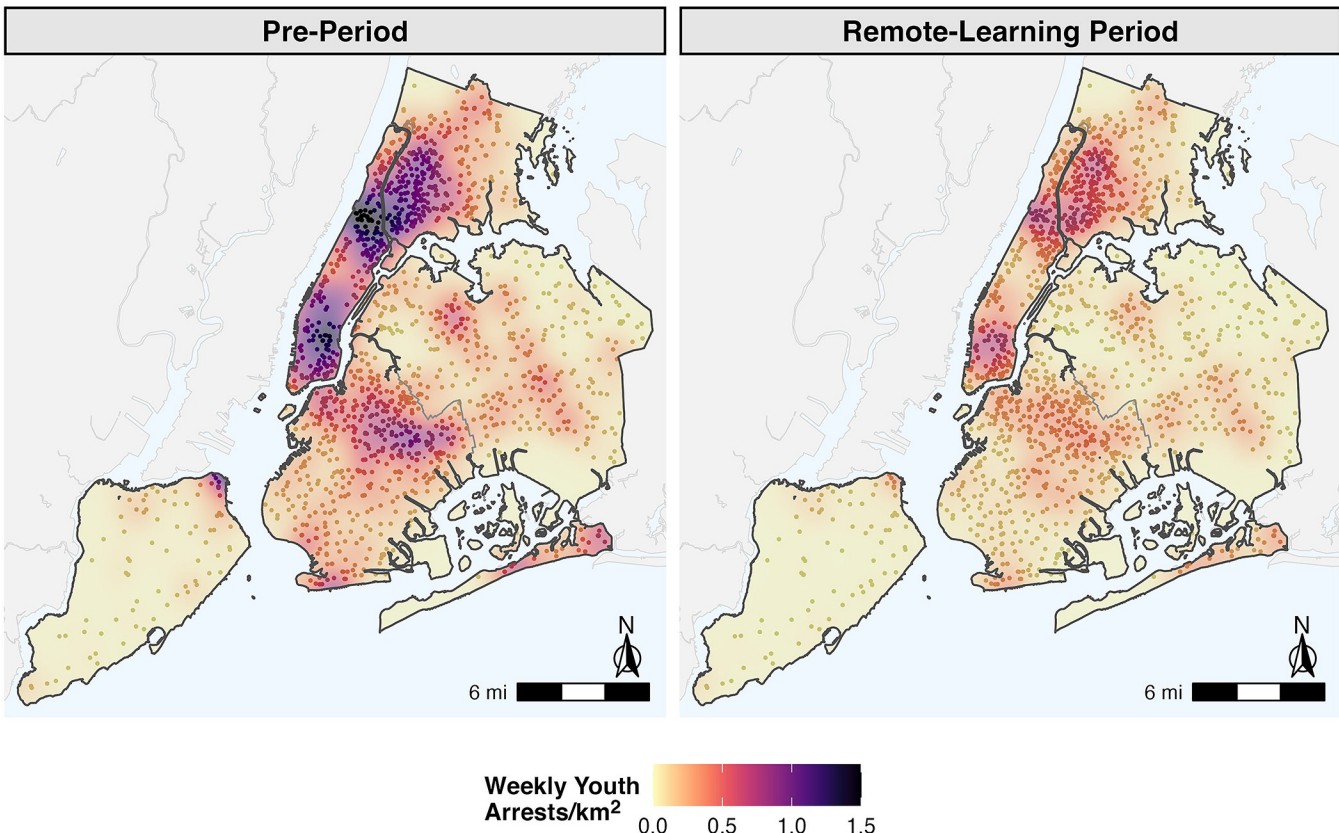

**Fig 4. Weekly youth arrest density (arrests/km$^2$) in school areas (300-foot buffer) and surrounding cities in pre- and post-remote learning periods, New York City, New York, 2019–2020.**

S6–S8 Figs). This map of New York City shows arrest density among youth declined during remote-learning periods, but remained highest in areas where youth arrest density was most concentrated in the pre-period.

Prior to and during periods of remote learning, 11.3% of arrests among youth occurred in the 300 feet surrounding schools compared to between 7.2 and 8.7% of arrests across all older age groups (Table 2). In addition, arrest density among youth in areas surrounding schools declined by 58.6% (95% CI: -63.9%, -53.0%), a greater percentage decline than any other age group (Table 2). Despite their comparatively smaller total population sizes, arrest densities in areas surrounding schools among Black and Latinx youth were at least 15 and 8.5 times that of White youth, respectively, in both pre- and remote-learning periods (Fig 5 and S2 Table). In

**Table 2. Weekly arrest density in school areas in pre- and remote learning periods (300-foot buffer).**

| Age (Years) | n Arrests | % Arrests in School Areas | Weekly Arrest Density in School Areas (Arrests/km2) | | | |
|---|---|---|---|---|---|---|
| | | | Pre-Period Density | Remote Learning Period Density | % Change Remote v. Pre | Abs. Change Remote v. Pre |
| <18 | 16,434 | 11.3% | 0.65 (0.62, 0.68) | 0.27 (0.24, 0.3) | -58.6% (-63.9%, -53.0%) | -0.38 (-0.43, -0.34) |
| 18–24 | 69,974 | 8.7% | 2.01 (1.97, 2.07) | 1.16 (1.09, 1.22) | -42.6% (-46.0%, -39.1%) | -0.86 (-0.94, -0.78) |
| 25–44 | 186,274 | 8.5% | 5.2 (5.11, 5.29) | 3.11 (3.01, 3.21) | -40.3% (-42.5%, -37.8%) | -2.09 (-2.24, -1.94) |
| 45–64 | 69,352 | 8.0% | 1.86 (1.81, 1.92) | 1.01 (0.95, 1.07) | -45.6% (-49.2%, -41.9%) | -0.85 (-0.92, -0.77) |
| 65+ | 4,740 | 7.2% | 0.12 (0.1, 0.13) | 0.06 (0.05, 0.08) | -46.4% (-59.7%, -29.2%) | -0.05 (-0.07, -0.03) |

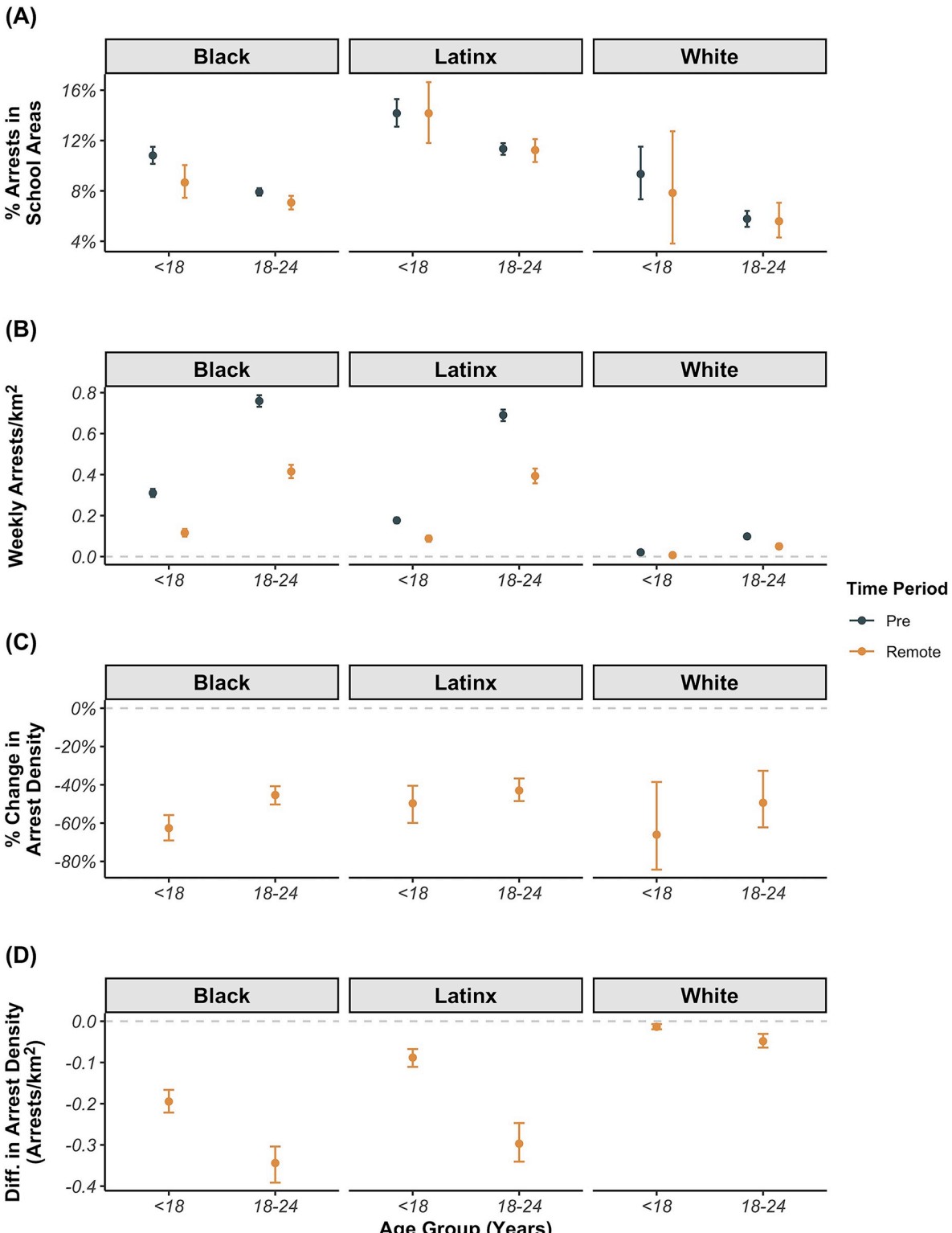

**Fig 5.** Percent of Arrests in School Areas (A), Weekly Density of Youth and Young Adult Arrests in school areas per km$^2$ (B), and Percent (C) and Absolute (D) Change in Arrest Density in School Areas by Time Period, Age Group, and Race/Ethnicity.

addition, a higher percentage of arrests among Black (10.3%) and Latinx (14.1%) youth occurred in areas surrounding schools compared with White youth (9.0%) (S2 Table). Among Black youth, the percentage of arrests in areas surrounding schools decreased from 10.6% in the pre-period to 8.6% during the remote-learning period (Fig 5). When the area surrounding schools was increased to 1000 feet, this inequity was even more pronounced. Results show 64.4% and 72.6% of arrests of Black and Latinx youth occurred in areas surrounding schools, respectively, compared to 50.3% among White youth (S2 Table).

Across all racial/ethnic groups, percentage declines in the density of youth arrests in areas surrounding schools during periods of remote learning compared to pre-periods were greater than observed in older age groups (Fig 5, S2 Table). Among youth, the magnitude of the percent decline in arrest density in areas surrounding schools was similar across racial/ethnic groups and ranged from -50.1% (-60.6%, -40.5%) among Latinx youth to -65.2% (95% CI: -84.1%, -38.4%) among White youth. While the magnitude of the percentage decline was similar across groups, the greatest absolute declines were observed among Black and Latinx youth. In sensitivity analyses, percentage declines in youth arrest density were slightly larger in areas surrounding schools compared to areas outside of school buffer zones [-58.6% (95% CI: -63.9%, -53.0%) vs. -53.0% (95% CI: -55.2%, -51.0%), respectively] (S9 Fig). This pattern was not observed among young adults, for whom percent declines in arrest density both within both school buffer areas and outside of school buffer areas were nearly identical (approximately -42.7%). These patterns were most pronounced among Black youth, for whom arrest densities declined by -62.4% (95% CI: -68.6%, -54.9%) in areas surrounding schools compared to -53.1% (95% CI: -55.3%, -50.1%) in areas outside of school buffer zones (S9 Fig).

## Discussion

Criminological research has largely focused on the ways in which schools may deter or incapacitate youth from engaging in criminalized behaviors [40–42]. Simultaneously, education inequality research has mainly studied the ways in which teachers, school officials, and school-based police officers criminalize youth [11, 13, 14]. Our results extend this body of work by examining a broader spatial structure linking youth arrests and school areas. Schools and their surrounding areas serve as aggregators for youth populations that are then specifically targeted by police [21, 43]. In this paper, we analyze what happens to youth arrest rates and racial/ethnic inequities when schools are unexpectedly closed for several months.

Analyzing police records from four urban cities to study the impacts of school closures on the arrest of youth, we find profound racial inequities in youth arrest rates throughout the study period (2019–2020). We find a significant decline in the arrest of all youth, and we observe the largest absolute declines for Black youth for whom average weekly arrest rates dropped by 26.8 per 100,000, versus 2.4 among White youth. These declines are consistent with recent data showing a decline in youth arrests through 2020 nationally [44]. However, despite these steep declines, Black youth arrests never approach rates among other racial or ethnic groups, before or after COVID-19 school closures. These dramatic reductions in youth criminalization after school closures in 2020 may have life-long socioeconomic and developmental implications. We find that pandemic-related school closures had a lasting effect on youth arrests as compared to young adults (age 18–24) through the immediate return to school period. Unlike other age groups, declines in youth arrests persisted even after partial school re-openings in Boston and Charleston, possibly due to hybrid learning, intermittent classroom and school closures, as well as pandemic-related changes in police and criminal court practices beyond schools [45]. For other age groups, arrests during Fall 2020 partial returns to in-person learning steadily rose back to levels before the start of the pandemic; in particular, young adult

arrests in New York City and Charleston were indistinguishable from pre-pandemic levels by the time schools partially re-opened.

Our spatial-temporal analysis of youth arrests and school location demonstrates the importance of looking beyond immediate school grounds for interrupting youth criminalization and disparities therein. Compared to older age groups, a higher proportion of youth arrests occurred in areas surrounding schools, especially arrests of Black and Latinx youth. In areas surrounding schools, we observed the greatest decline in arrest density for youth compared to older age groups both overall and for specific racial/ethnic groups. While the magnitude of the percentage decline was similar across race/ethnicity groups, the greatest absolute decline in arrests in school areas was observed among Black and Latinx youth, and Black youth saw a significantly lower percentage of arrests occurring in areas surrounding schools as compared to Black young adults. In sum, the data suggest school closures deemphasized schools for police activity, which constitute a main way youth encounter police. Mapping and spatial analysis show that school zones comprise a significant portion of urban space depending on their size (from 300 to 2650 feet), the size and density of the city, its population, and the number of density of its public institutions. Under these conditions, youth are particularly vulnerable to the spatial conditions of policing and sentencing enhancement. More broadly, we aim to contribute to a growing body of research that examines the spatial context of policing [46] and argue that school locations are an important driver to youth criminalization. School closures thus significantly limited youth contact with police officers, who may have been arresting youth in systematic ways through policy and practice, such as the surveillance of school areas and school property. However, school closures are not a policy solution to the social problem of youth criminalization, rather, these results point to the need for policy interventions addressing how policing is enacted in school areas.

School districts are beginning to address school-based criminalization; the American Rescue Plan provided resources invest in alternatives to youth criminalization, and in 2020 at least 33 school districts ended the use of school resource officers [47]. Although it is important to address and draw attention to the increased role of police officers on school grounds, findings from this paper support a broad strategy of reform that considers the full spectrum of social contexts of youth—including home neighborhoods, modes of transportation to and from school, and public parks—as multiple leverage points for policies addressing the policing of youth [10, 15, 18]. Moreover, states can also set a minimum age for criminal prosecution (ranging from age 6–12), but 28 states do not have a minimum age law, including South Carolina—a state we examine in the current study [48].

Research on the impacts of school closures during COVID-19 has weighed the negative effects for youth alongside the public health importance of these policies for minimizing morbidity and mortality. For overly surveilled and underserved students, school closures meant a lack of access to resources for health needs and a disruption in school routines that are important coping mechanisms for young people [49, 50]. Alongside other pandemic-related stressors, including parental/caregiver bereavement, school closures may have had widespread negative impacts for youth, including increased food insecurity, extensive learning loss, and exacerbated social inequities [51–56]. Our research contributes to this literature by suggesting that in addition to significant reductions in mortality [57], school closures may have also had a positive impact on reducing youth's exposure to the criminal legal system, especially for Black and Latinx youth. These findings should be contextualized within the broader literature on the health consequences of frequent and unequal contact with police among youth [2–4, 58, 59]. The precipitous decline in arrests for all youth, but especially for Black youth, could have been a broad public health benefit and may reduce racial health inequities.

Our analysis is subject to limitations, and future inquiry could expand the current study's findings. First, we cannot account for changes that may have occurred not solely due to school closures, but stay-at-home ordinances as well. However, we also observed declines in youth arrests during the summer weeks of 2019, several months before the onset of the COVID-19 pandemic. Moreover, our spatial models indicate that many of the declines among youth in the areas surrounding schools during the pandemic were not as pronounced for other age groups who were also subject to pandemic-related changes in policing practices and stay-at-home ordinances. We cannot distinguish changes in arrest due to policing behavior or youth population behavior; we can only observe that an arrest occurred. Future studies could examine arrests during other kinds of school closures unrelated to the pandemic, such as teacher strikes or summer/winter breaks, or examine specific time windows of the school week or weekend [60]. Second, the spatial density measure cannot be informative about whether a youth attended that school, precluding direct tests of in-school arrests. Future research could collect individual-level data from schools and perform record linkages with criminal record information to study patterns of criminalization emanating from school attendance. Furthermore, future research should explore the potential that neighborhood context, including levels of segregation, poverty, concentrated disadvantage, or racial composition, could significantly moderate the relationship between school location and youth arrest rates. Indeed, we believe our results suggest the joint spatial context of schools and neighborhoods should be examined to better understand youth criminalization. Finally, although we use arrest data from 2019 as a point of temporal comparison, due to data availability across multiple jurisdictions, our study was unable to include additional years of data with which to understand these trends.

## Conclusions

Taken together, these results challenge existing research and received wisdom about the relationship between youth, schools, and the spatial-temporal pattern of criminalized behaviors. Our findings suggest that reducing police contact in childhood and adolescence, especially for Black and Latinx youth subject to the unequal effects of structural racism, will require that policymakers consider a broad set of policy actions spanning educational and criminal justice institutions. Hot spots policing and other targeted approaches around schools may significantly increase the risk of criminalization in childhood. These practices, designed to target crime problem areas, should be evaluated in light of recent research using a natural experiment that finds little evidence schools generate higher rates of neighborhood crime [23]. Our findings suggest that the assumed link between crime and school location is likely overstated and may be attributable in part to increased enforcement and surveillance practices around schools. Our analysis demonstrates the ways law enforcement alters practices with respect to space (i.e., school location) and time (i.e., summer recess), decisions which are highly influential for the risk of police arrest among youth of color. Research and policy focused on out-of-school time as a risk factor for criminalized youth behaviors should also consider how schools being in regular session will structure policy enforcement activity. In sum, a comprehensive approach to addressing youth criminalization and disparities should consider policing strategies such as hot spots, directed patrol, and other place-based police interventions near school grounds as a lever for reducing the prevalence of youth arrests, especially for Black and Latinx youth facing high risk of criminalization.

The COVID-19 pandemic has laid bare the impacts of structural racism that predate the current crisis. The large racial inequities in arrest of youth documented in this study are the result of inequitable policies that remain unaddressed. The pandemic posed significant challenges, but through the mechanism of school closure, it reduced the arrest of Black youth by

55% after school closed and attenuated the relationship between Black and Latinx youth arrests and the geography of K-12 public schools. As schools have fully reopened to in-person learning, cities have an opportunity to reevaluate the use of policing within and surrounding schools and the use of arrest among youth populations.

## Supporting information

**S1 Table. Negative binomial regression results of arrests by age group before and after school closures, 2019–2020.**
(DOCX)

**S2 Table. Percentage of arrests in school areas and arrest density in school areas pre- and post-remote learning periods by age group and race/ethnicity (300-foot buffer).**
(DOCX)

**S3 Table. Sensitivity analysis for percentage of arrests in school zones and arrest density in school areas pre- and post-remote learning periods by age group and race/ethnicity using 1,000-foot and 2,640- foot buffer zones.**
(DOCX)

**S4 Table. Percentage of city land area covered and percentage of youth and young adult arrests occurring in using buffer sizes of 300-ft (main analysis), 1,000-ft and 2,640-ft buffer zones, overall and by city.**
(DOCX)

**S1 Fig. Temporal trends in arrest rates by age group in in Boston, Charleston, Pittsburgh, and New York City (2019–2020).**
(DOCX)

**S2 Fig. Change in weekly youth arrests rates after school closures as compared with arrests during summer 2019.**
(DOCX)

**S3 Fig. Change in weekly youth arrest rates before vs. after school closures in Boston, Charleston, Pittsburgh, and New York City (2019–2020).**
(DOCX)

**S4 Fig. Change in weekly youth arrest rates before vs. after school closures by race and ethnicity (2019–2020).**
(DOCX)

**S5 Fig. Spatial extent of different school buffer zone distances used in main (300-ft) and sensitivity analyses (1,000-ft and 2,640-ft).**
(DOCX)

**S6 Fig. Weekly youth arrest density (arrests/km2) in school areas (300-foot buffer) and surrounding cities in pre- and post-remote learning periods, Boston, Massachusetts.**
(DOCX)

**S7 Fig. Weekly youth arrest density (arrests/km2) in school areas (300-foot buffer) and surrounding cities in pre- and post-remote learning periods, Charleston, South Carolina.**
(DOCX)

**S8 Fig. Weekly youth arrest density (arrests/km2) in school areas (300-foot buffer) and surrounding cities in pre- and post-remote learning periods, Pittsburgh, Pennsylvania.** (DOCX)

**S9 Fig. Percent change in arrest density for youth and young adult arrests within school buffer areas (300-feet) and outside of school buffer areas, overall and by race/ethnicity.** (DOCX)

## Acknowledgments

We thank Jonathan J.B. Mijs, Jonathan S. Jay, Catherine d.P. Duarte, Max Greenberg, Brenden Beck, Caylin Louis Moore, and Claudia Anderson for helpful comments on the manuscript.

## Author Contributions

**Conceptualization:** Jessica T. Simes, Tori L. Cowger.

**Data curation:** Jessica T. Simes, Tori L. Cowger, Jaquelyn L. Jahn.

**Formal analysis:** Jessica T. Simes, Tori L. Cowger, Jaquelyn L. Jahn.

**Funding acquisition:** Jessica T. Simes, Jaquelyn L. Jahn.

**Investigation:** Jessica T. Simes, Tori L. Cowger, Jaquelyn L. Jahn.

**Methodology:** Jessica T. Simes, Tori L. Cowger, Jaquelyn L. Jahn.

**Project administration:** Jessica T. Simes, Tori L. Cowger, Jaquelyn L. Jahn.

**Resources:** Jessica T. Simes.

**Visualization:** Jessica T. Simes, Tori L. Cowger, Jaquelyn L. Jahn.

**Writing – original draft:** Jessica T. Simes, Tori L. Cowger, Jaquelyn L. Jahn.

**Writing – review & editing:** Jessica T. Simes, Tori L. Cowger, Jaquelyn L. Jahn.

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
