## [Decision Letter · Decision Letter 0]

6 Mar 2023

PONE-D-22-31399School Closures Significantly Reduced Arrests of Black and Latinx Urban YouthPLOS ONE

Dear Dr. Jahn,

Thank you for submitting your manuscript to PLOS ONE.  After careful consideration, we feel that it has merit but does not fully meet PLOS ONE’s publication criteria as it currently stands.  Therefore, we invite you to submit a revised version of the manuscript that addresses the points raised during the review process. I have received a report from an expert reviewer in criminology who enjoyed the paper but recommends revisions.  I concur with this reviewer's opinion and have also reviewed your paper, serving as Reviewer 2.  Reviewer 1's comments can be viewed at the bottom of this email.  I also have attached Reviewer 1's comments and my report to this email as separate files.   

We look forward to receiving your revised manuscript.

Kind regards,

W. David Allen

Academic Editor

PLOS ONE

Journal Requirements:

3**. **Please include your tables as part of your main manuscript and remove the individual files. Please note that supplementary tables (should remain/ be uploaded) as separate "supporting information" files

Reviewers' comments:

Reviewer's Responses to Questions

**Comments to the Author**

1. Is the manuscript technically sound, and do the data support the conclusions?

Reviewer #1: Yes

2. Has the statistical analysis been performed appropriately and rigorously? 

Reviewer #1: Yes

3. Have the authors made all data underlying the findings in their manuscript fully available?

Reviewer #1: Yes

4. Is the manuscript presented in an intelligible fashion and written in standard English?

Reviewer #1: Yes

5. Review Comments to the Author

Reviewer #1: The study capitalizes on a natural experiment resulting from school shutdowns due to the COVID-19 pandemic to measure the impact of policing schools as “hot spots” on racial disparities of youth criminalization. Results are clear and compelling and have important implications for policy. I offer the following comments to potentially improve the manuscript.

Methods:

Multiple comparisons are made to support the claim that areas around schools are particularly important for the criminalization of Black and Latinx youth—comparisons between pre- and post-school closures, comparisons by race, comparisons by city, and comparisons by age group. Why not also compare arrest density of school buffers zones with arrest density outside of school zones? It would seem to be the most direct comparison to support the overall conclusion.

Also, the fixed effects approach is effective for controlling for unmeasured covariates that differ between cities (and season). What remains unexamined here are the potential confounders (or moderators!) due to within-city inequality based on variables that have been tested extensively in criminological research such as social disorganization, resource disadvantage, median income, percent HS grads, % aged 15-24, population density, etc. Indeed, Percent Black and Percent Latinx might even be most relevant here. That said, I think it’s reasonable to argue that an extensive analysis is beyond the current scope, but a supplemental analysis to check for variation, say between school sites within cities, would be a helpful clarification (and/or signpost for future research).

Please clarify: Do arrest rates within the buffer zones include in-school arrests? How does this impact findings?

Results:

The text on pg. 12 (Youth Arrest Rates and Disparities…) combined with Figure 1 is unclear. Is the cited rate of average weekly youth arrest rates of 17.4/100,000 for 2019 meant to represent the average for the whole year? Does that include Jan-Mar 2020? Or does it represent only Jan-Mar 2020 rather than 2019? Also, perhaps I am not understanding what the trendlines represent, but why does the trajectory at the end of 2019 not line up with the beginning of 2020? Most importantly—from Jan-Mar 2020, the trajectory already appears to be heading downward. What could explain it? How do we rule out that the levels after Mar 2020 are not simply a continuation of processes that started in Jan, rather than the school shutdowns?

Also – please define IRR at some point in the study.

Very minor:

Pg. 7, line 106: I appreciate the magnitude of the impact of these findings on health outcomes, but I actually think the authors might be selling their reach here a bit short, as the findings also have implications for economic outcomes (e.g., Pager 2003 – The Mark of Criminal Record), not to mention education, and future criminal justice contact as well.

6. PLOS authors have the option to publish the peer review history of their article (what does this mean?). If published, this will include your full peer review and any attached files.

Reviewer #1: No

---

## [Author Response · Author response to Decision Letter 0]

24 May 2023

** Please see files at the end of this PDF for Author's Response to Reviewers with Figures**

Response to Review of 

“School Closures Significantly Reduced Arrests of Black and Latinx Urban Youth”

PONE-D-22-31399

We are very grateful to the reviewer and editor for their comments on the paper. We have substantially revised the manuscript and tried to respond to all concerns. Below is a summary of the major revisions and a point-by-point response to Reviewers 1 and 2. Original reviewer comments are in blue.

Summary of Major Revisions

• Methods: We provide more details on the methodology, including an equation, and we explain that we are using negative binominal regression analysis to model counts of arrests (with a city population offset) due to overdispersion in the dependent variable.

• Results and Sensitivity Analyses: We provide clarification on our main findings reported in Figure 1. We provide a new set of maps in the main paper and supporting information (SI) to demonstrate the spatial context of school buffer areas and arrests. Finally, an additional sensitivity analysis examines changes to arrests for youth and young adults by race/ethnicity, comparing arrest densities inside and outside of school buffer areas. 

• Framing and Discussion: We have provided additional motivation and clarification of the study’s limitations and implications in both the Introduction and Discussion sections.

Reviewer 1

Point 1. Multiple comparisons are made to support the claim that areas around schools are particularly important for the criminalization of Black and Latinx youth—comparisons between pre- and post-school closures, comparisons by race, comparisons by city, and comparisons by age group. Why not also compare arrest density of school buffers zones with arrest density outside of school zones? It would seem to be the most direct comparison to support the overall conclusion.

We appreciate this comment and agree it is a useful and important addition to the current paper given our focus on arrest rates by school zone area. We have substantially revised the manuscript to respond to this comment. We describe a new sensitivity analysis on p. 12 of the revised manuscript (lines 262-64). We report the results of this sensitivity analysis on p. 17 (lines 519-526), where we find percentage declines in youth arrests were larger in surrounding school areas compared to areas outside of school buffer zones. While this pattern was most pronounced among Black youth (arrest declines were largest for Black youth within school buffers), confidence intervals for results within and outside of school buffer zones overlap. However, these results lend support for our main findings in Fig 5.

Point 2. Also, the fixed effects approach is effective for controlling for unmeasured covariates that differ between cities (and season). What remains unexamined here are the potential confounders (or moderators!) due to within-city inequality based on variables that have been tested extensively in criminological research such as social disorganization, resource disadvantage, median income, percent HS grads, % aged 15-24, population density, etc. Indeed, Percent Black and Percent Latinx might even be most relevant here. That said, I think it’s reasonable to argue that an extensive analysis is beyond the current scope, but a supplemental analysis to check for variation, say between school sites within cities, would be a helpful clarification (and/or signpost for future research).

We appreciate this comment, and we agree that there is the potential for within-city effect measure modification/moderation. In our paper, we examine two main moderators in the changes in arrest: (1) arrested person’s age and race/ethnicity; (2) school buffer zones. Declines in arrests within school areas may have been more substantial in certain neighborhoods within cities compared to others, and we think future research could examine whether arrest declines were more or less substantial in school zones that overlap with greater socioeconomic disadvantage or higher levels of segregation. In our discussion of limitations and future research, we now say that future work should assess neighborhood context as a moderator of the school-policing relationship (p. 21, lines 636-641).

In terms of confounding, the variables described are largely static over our study period, which means that our fixed effect approach addresses these potential confounders since they are largely invariant. For example, we do not expect the proportion of the Black population to have dramatically changed over the course of the first year of the pandemic.

Point 3. Please clarify: Do arrest rates within the buffer zones include in-school arrests? How does this impact findings?

Arrests within buffer zones include any arrest that took place within the given buffer area; this includes arrests on the school site, as well as beyond it (but within the buffer). We clarify this on page 11, line 232-233 in the revised manuscript. Because the study is interested in how city police (as opposed to school district officers) interact with school location before and after school closures, we think it is appropriate to include any arrest recorded by city police that took place within a buffer zone, within or outside of school grounds.

Point 4. The text on pg. 12 (Youth Arrest Rates and Disparities…) combined with Figure 1 is unclear. Is the cited rate of average weekly youth arrest rates of 17.4/100,000 for 2019 meant to represent the average for the whole year? Does that include Jan-Mar 2020? Or does it represent only Jan-Mar 2020 rather than 2019? 

We clarify which weeks comprise the reported average (17.4 per 100,000) during the pre-period on page 13 (lines 299-300) to be all weeks in 2019 up to state-specific school closures in mid-March.

Point 5. Also, perhaps I am not understanding what the trendlines represent, but why does the trajectory at the end of 2019 not line up with the beginning of 2020? Most importantly—from Jan-Mar 2020, the trajectory already appears to be heading downward. What could explain it? How do we rule out that the levels after Mar 2020 are not simply a continuation of processes that started in Jan, rather than the school shutdowns?

We appreciate this comment, and we think there are two reasons for this. First, arrests can vary greatly week-to-week, so we use a loess line with bin size .5 to reduce noise in the scatterplot. Second, as early reports of COVID-19 cases began in the early days of March, small changes to arrests in the weeks prior to school closure may be driving the change in the curve prior to school closure. However, these chances are minor, and below we include the same plot, with a smaller bin size (.1), which allows for the loess curve to be more influenced by the data points. For clarity and consistency across Figures 1 and 3 (and S1 Fig), we have reduced the bin size for the loess function.

Memo Figure 1. Replicating Figure 1 with bin size .1

Point 6. Also – please define IRR at some point in the study.

IRR is now defined on page 10 of the revised manuscript.

Point 7. Pg. 7, line 106: I appreciate the magnitude of the impact of these findings on health outcomes, but I actually think the authors might be selling their reach here a bit short, as the findings also have implications for economic outcomes (e.g., Pager 2003 – The Mark of Criminal Record), not to mention education, and future criminal justice contact as well.

We now expand our discussion of the implications of the study to reflect this point, stating on page 7, lines 131-133 of the revised manuscript: “Declines in arrests may also have implications for youth social and economic outcomes, including education, employment, and future criminal justice contact,” and cite Devah Pager’s 2003 article.

Reviewer 2

Point 1. I think you can emphasize your research strategy somewhat more forcefully or prominently when motivating your paper and its attributes, especially in the introduction.

We appreciate this comment and have revised the introduction on page 6, lines 117-118: “We use 2020 COVID-19 pandemic school closures as a novel exogenous interruption to in-school attendance and thus potentially youth-police contact in school areas.”

Point 2: The introduction of the paper should speak more pointedly to the question of why, as scholars or as a society, we should care about “harm” to youths arrested for committing actual offenses? One could argue that they have earned their harm, and it may not be appropriate to expect society to care about that harm. A passage on page 5 illustrates what I mean a little more clearly. There, you assert that schools embedded in neighborhoods create “criminal justice contact” that leads to criminalization of youth of color, thus affecting (presumably adversely) their “orientation towards both schooling and the criminal justice system.” I can see how this would be a problem when youth of color are targeted by police for illegalities they did not commit or for the same (legal) activities undertaken by white kids, the theme of the example you cite later in that paragraph. But is there evidence that youth of color are not committing actual offenses (disproportionately)? The point is that at least some of the criminalization of youth of color that takes place in the context of neighborhood schools as policing hot spots might just be an artifact of race-criminality patterns that exist, independent of racially biased policing. Put another way, some youth (those who actually commit offenses) criminalize themselves by the choices they make; they are, after all, youths, and young people do make mistakes. (We all support actions that help them make better choices, both in general and in the aftermath of an arrest, but of course your paper is principally concerned with the actions of police, and that’s fine.) Ultimately I think you could improve this passage, and thus the introduction overall, by being clearer on what the true problem actually is, and what it isn’t, in the empirical setting you study here.

We very much appreciated the opportunity to reflect on this set of comments. We have two main responses. First, in our view, the culpability of youths for their arrest charges, or the deservedness of one’s arrest more generally, is not the central focus of the analysis, nor can we assess youths’ behavior, legal or not, in this analysis. We thus analyze arrest data as merely an instance of police arrest, making no assumptions about the underlying behavior of the youths. Second, there is a large body of evidence, which we cite throughout the paper, indicating that arrests in childhood and adolescence can lead to significant negative consequences across a variety of domains—social, economic, and health. Given this evidence, we would argue for greater scientific attention to alternatives to policing and incarceration of youth, as one future direction from this study. We believe there are alternative ways to respond to criminalized behavior among children and adolescents, and the harshest outcomes (arrest and incarceration) should be used with extreme parsimony, particularly for groups harmed by structural racism and other systems of marginalization.

We agree that this passage on page 5 can be clarified to state the problem our analysis addresses is, and the empirical setting we study. We have revised the text on p. 5 (lines 82-83) to reflect this discussion, and we thank the reviewer for pushing us to be clearer. Finally, after considering these comments, we decided our characterization of the arrests as an artifact of “hot spots policing” was too narrow and instead should be talked in broader terms. We have adjusted the language around policing strategies throughout the manuscript to reflect this change in framing.

Point 3. At the end of the short section on sensitivity analysis, you note, as part of your spatial analyses, that you considered school zone buffer distances of 1,000 feet for New York and Pennsylvania (it would be better if you said Pittsburgh here) and 2,640 feet for South Carolina (it would be better if you said Charleston here). I think this passage could be improved if you could give us a general feel for the context of these distances in practice and why you focus on two different distances for the two sets of cities. For example, within 1,000 feet of a typical (defined however you like) school in New York or Pittsburgh, is the landscape dominated by residences, or is it far enough away that you see no residences but instead a significant presence of stores or other buildings and establishments? Might there be a mix of these? Such detail would give the reader a clearer understanding of the lay of the land of the school neighborhoods that you’re data cover.

We appreciated this comment, and we thought that maps would be the best way to convey information about the spatial context of school buffer areas and arrest densities. We have substantially revised the manuscript to include a series of maps that provide readers with some context for the distances in practice, and the implications for the different buffer zones. For the main paper, we focus on New York City, because the data from NYC covers approximately 90% of arrests in the data. We provide additional maps in the Supplemental Information for other cities. We first provide maps of the three buffer areas we consider in S5 Figure in the Supplemental Information (see discussion on page 11, lines 235-238 of the revised manuscript). Figure 4 in the main paper displays arrests before and after remote learning was implemented in New York City, with school buffer densities drawn around school locations. We believe these maps add greater context to the spatial areas and distances in practice. Finally, although school zones are statewide policies, we recognize that the data only contain these cities, so we have also made edits to the sensitivity analysis to only refer to cities, not states.

Point 4. Right now your paper is rather sparse on econometric details. When introducing your negative binomial regression model(s), you should show your regression equation in detail, making clear how your focus variable enters the equation and how its effect will be estimated, so that the reader understands the model’s essential structure. It would also be helpful to remind the reader that you have count data, which motivates NB regression in the first place, and to comment on why you are not estimating Poisson models. Presumably, your data exhibit significant overdispersion that makes Poisson regression invalid statistically, which can be readily tested; document and comment on the results of this test. Finally, accompanying these revisions, you would do well to show a table (or tables) showing the full results of your NB regression results, not just summaries of the focus results.

We have made several changes to the manuscript to respond to this point. First, we provide a regression equation on page 10 of the revised manuscript. Here, we also add text that reports the dependent variable as a count explaining that we are using negative binominal regression analysis to model counts of arrests (with a city population offset) due to overdispersion in the dependent variable. We report the dispersion parameter for our main model indicated significant overdispersion in weekly arrests (10.321, taken to be 1), and AIC tests indicate NB is preferable to Poisson. We also provide a full table of results in the SI for the modeled results reported in Fig 2 (S1 Table). For regression results only reported in the SI, we have now noted that full regression results available upon request.

Point 5. Although exploiting systematic school closures functions as an interesting identification strategy for your essential research question—helping you get at the effect of a policy-based, exogenous reduction in youth-police interaction in school neighborhoods—the closing of schools cannot be the lasting policy recommendation, as this is just not practical. Your paper needs to acknowledge this and emphasize that ultimately the lasting implication concerns the nature of policing in those neighborhoods.

We agree with this point and have added the following passage to the discussion section (page 20, lines 597-599): “However, school closures are not a policy solution to the social problem of youth criminalization, rather, these results point to the need for policy interventions addressing how policing is enacted in school areas.”

Point 6. Within the Discussion section, you note that a number of school districts have recently ended the use of school resource officers, which then draws attention to the role of police officers. I suspect many readers do not know this. You might find it interesting to tap into a recent study of mine, titled “Self-Protection Against Crime: What Do Schools Do?” (Applied Economics 2018). It’s possible that its findings, based on an economic analysis of guard use inside schools, could lend perspective to your closing discussion with respect to how you think about the problem you are studying and its empirical setting. Please note, however, that this is just a suggestion and by no means a requirement for your revision.

We appreciate this suggestion and, respectfully, chose to decline this suggestion. Our paper’s introduction discusses previous research on school resource officers (SRO) to show that SROs has been a primary focus of previous literature on youth arrests at school. Our paper’s contribution is its broader spatial focus on the areas surrounding schools, and we chose to focus the discussion accordingly: “Although it is important to address and draw attention to the increased role of police officers on school grounds, findings from this paper support a broad strategy of reform that considers the full spectrum of social contexts of youth—including home neighborhoods, modes of transportation to and from school, and public parks—as multiple leverage points for policies addressing the policing of youth” (Pages 20, lines 602-606).

---

## [Editor Report · Decision Letter 1]

12 Jun 2023

School Closures Significantly Reduced Arrests of Black and Latinx Urban Youth

PONE-D-22-31399R1

Dear Dr. Jahn,

Thank you for submitting your revised manuscript.  I very much appreciate the thoroughness with which you revised your paper and documented those revisions.  You and your co-authors have produced a very interesting study. 

We are pleased to inform you that your manuscript has been judged scientifically suitable for publication and will be formally accepted for publication once it meets all outstanding technical requirements.

Kind regards,

W. David Allen

Academic Editor

*PLOS ONE*
---

## [Editor Report · Acceptance letter]

28 Jun 2023

PONE-D-22-31399R1 

School Closures Significantly Reduced Arrests of Black and Latinx Urban Youth 

Dear Dr. Jahn:

I'm pleased to inform you that your manuscript has been deemed suitable for publication in PLOS ONE. Congratulations! Your manuscript is now with our production department. 

Kind regards, 

on behalf of

Dr. W. David Allen 

Academic Editor

PLOS ONE